# A Recurrent Neural Circuit Mechanism of Temporal-scaling Equivariant Representation

**Junfeng Zuo**[1]
zuojunfeng@pku.edu.cn

**Xiao Liu**[1]
xiaoliu23@pku.edu.cn

**Ying Nian Wu**[2]
ywu@stat.ucla.edu

**Si Wu**[1]
siwu@pku.edu.cn

**Wen-Hao Zhang**[3,4*]
wenhao.zhang@utsouthwestern.edu

[1]Peking-Tsinghua Center for Life Sciences, Academy for Advanced Interdisciplinary Studies,
School of Psychological and Cognitive Sciences,
Beijing Key Laboratory of Behavior and Mental Health,
IDG/McGovern Institute for Brain Research,
Center of Quantitative Biology, Peking University.
[2]Department of Statistics, University of California, Los Angeles.
[3]Lyda Hill Department of Bioinformatics, UT Southwestern Medical Center.
[4]O'Donnell Brain Institute, UT Southwestern Medical Center.

## Abstract

Time perception is fundamental in our daily life. An important feature of time perception is temporal scaling (TS): the ability to generate temporal sequences (e.g., movements) with different speeds. However, it is largely unknown about the mathematical principle underlying TS in the brain. The present theoretical study investigates temporal scaling from the Lie group point of view. We propose a canonical nonlinear recurrent circuit dynamics, modeled as a continuous attractor network, whose neuronal population responses embed a temporal sequence that is TS equivariant. We find the TS group operators can be explicitly represented by a time-invariant control input to the network, whereby the input gain determines the TS factor (group parameter), and the spatial offset between the control input and the network state on the continuous attractor manifold gives rise to the generator of the Lie group. The recurrent circuit's neuronal responses are consistent with experimental data. The recurrent circuit can drive a feedforward circuit to generate complex sequences with different temporal scales, even in the case of negative temporal scaling ("time reversal"). Our work for the first time analytically links the abstract temporal scaling group and concrete neural circuit dynamics.

## 1 Introduction

We are living in a dynamic world and the brain can flexibly process sensory and motor events occurring at different time scales [1–4]. An example of such temporal flexibility is self-initiated movements, e.g., singing a song at normal or faster speed. The capability of temporally flexible movements implies the brain flexibly control the underlying neural circuit dynamics. Indeed, experimental data converges to an empirical principle of temporal scaling (TS) at the behavioral and neuronal levels [2–5]. Specifically, when generating movements with longer intervals, it was found the neuronal population activities evolve along the same (low-dimensional) manifold of the neuronal population

---

*Corresponding author.

37th Conference on Neural Information Processing Systems (NeurIPS 2023).

responses but at a slower speed [2–8]. Although this observation was reproduced in previous modeling studies that trained recurrent circuit models to achieve temporal scaling (e.g., [2, 6, 7, 9–15]), it is far from clear about the mathematical principle governing temporal scaling in brain's recurrent neural circuits.

Temporal scaling is also fundamental in machine learning (ML) and robotic research. Most of engineering approaches use time warping to achieve temporal scaling where sinusoidal oscillatory inputs are needed (e.g., [16–19]). However, no experiments support such oscillation-based signals are used to represent time information ranging from milliseconds to seconds in the brain [3–5]. Some other ML studies directly modulated time constants of artificial neurons in recurrent neural networks to realize TS (e.g., [20]), whereas real neurons' time constant is thought to be fixed due to biophysical properties. Earlier studies also introduced neurophysiological properties to achieve temporal scaling, e.g., using synaptic shunting to adapt the effective integrating time [21] and utilizing time-covariant time field sizes [22]. However, they ([21, 22]) focused on the scale invariance during the temporal scaling inference rather than generating temporal sequences with different speeds. Combined, current ML models adopt different mechanisms to realize TS than neural circuits in the brain.

To provide algebraic and dynamical understanding of temporal scaling in neural circuits, we rigorously study this question from temporal scaling (Lie) group. We contend that there are two aspects of realizing TS group transformations in recurrent neural circuits: One is *explicitly* representing a TS controller corresponding to TS group operators; and the other is representing a TS equivariant temporal sequence in spatiotemporal neuronal responses, referred to as TS *equivariant* representation. It is unknown how recurrent neural circuits represent abstract TS group operators, nor the equivariant representation of temporal sequences. And no previous ML study investigated the equivariance of a Lie group acting on the temporal domain, nor an explicit representation of group operators.

The present study *analytically* derives a canonical current circuit with continuous manifolds of attractors, so-called continuous attractor networks (CANs), to equivariantly represent generic sequences with different temporal scales. And the TS operators are realized by a *time-invariant* control input applied to the CAN: The TS factor is represented as the gain (magnitude) of the control input, and the TS (group) *generator* emerges from the spatial offset between the control input and the network state along the continuous attractor manifolds. Moreover, applying a negative (inhibitory) gain of control input enables the recurrent circuit to generate a *time-reversed* sequence. The proposed circuit model reproduces a wide range of experimental findings of temporal scaling. We also demonstrate the recurrent circuit can drive a feedforward circuit to generate arbitrary, complex temporal sequences with different time scales, e.g., hand-written sequences. It will shed light on understanding the flexible timing in the brain, and provides new building blocks to achieve timing tasks in engineering.

The contributions of our work are as follows: It for the first time analytically links the TS group equivariant representation to a biologically plausible recurrent neural circuit, and analytically identifies that TS group operators are represented by time-invariant control inputs applied to the circuit. The proposed circuit with *fixed connectivity* flexibly generates TS equivariant sequences by just modulating the gain of control inputs, which is a novel mechanism never used in ML studies (Discussion, comparison with other works). It also proposes a novel circuit mechanism generating time-flipped sequences ("time reversal") with receiving control inputs of negative gain.

## 2 The temporal scaling group

We formulate the temporal scaling transformation by using the Lie group. Consider a TS group $\mathbb{S}$ whose group operator $S(\alpha) \in \mathbb{S}$ will map a time scale $t$ (the scale of the time axis) into a new scale $\mathbb{t}$,

$$t \mapsto \mathbb{t} = S(\alpha) \cdot t = t_{\text{start}} + \alpha t, \tag{1}$$

where the symbol $\cdot$ denotes the action of group operators, and $\alpha$ is the temporal scaling factor. $t_{\text{start}}$ denotes the start timing, where the $t_{\text{start}} = t_0 = 0$ when $\alpha \geq 0$, and otherwise $t_{\text{start}} = t_\infty$ regards as the end timing of the original time scale. Across this study, we use $t$ to represent a "standard" or reference physical time, and $\mathbb{t}$ denotes the scaled time. Then we consider an arbitrary temporal sequence $\mathbf{y}(t)$, which can be high-dimensional and may be regarded as, e.g., hand movements, vocal sequences, etc. Changing the time scale $t$ will transform $\mathbf{y}(t)$ into a new sequence $\mathbf{y}(\mathbb{t})$,

$$\mathbf{y}(\mathbb{t}) = \hat{S}(\alpha) \cdot \mathbf{y}(t) = \mathbf{y}[S(\alpha) \cdot t] = \mathbf{y}(\alpha t), \tag{2}$$

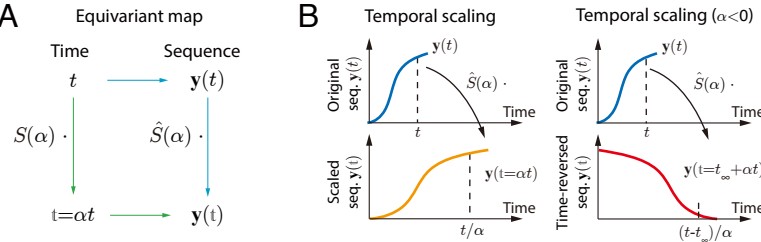

Figure 1: (A) An equivariant map between the time and a temporal sequence. (B) Left: A TS operator $\hat{S}(\alpha)$ with $\alpha > 0$ scales the time by a factor $\alpha$ and then scale temporal sequences accordingly. Right: A TS operator $\hat{S}(\alpha)$ with $\alpha < 0$ flips and scales the sequence on time axis.

To simplify notation, we suppress $t_{\text{start}}$ in the last term in Eq. (2) and across the study, in that our focus is the temporal scaling of sequences rather than temporal alignment between them. Our actual calculations do align $t_{\text{start}}$ appropriately. $\hat{S}(\alpha)$ is a TS operator acting on a sequence, in contrast to $S(\alpha)$ acting on the time $t$ directly. TS group operators are *commutative*, i.e., the effect of sequential actions of two operators can be composed into the same operator irrelevant to their order, i.e., $\hat{S}(\alpha) \cdot \hat{S}(\beta) = \hat{S}(\beta) \cdot \hat{S}(\alpha) = \hat{S}(\alpha\beta)$. Other properties of TS group operators can be found in Supplementary Information (SI) (Eq. S1).

$\alpha > 1$ $(0 < \alpha < 1)$ corresponds to speed up (slow down) the time, analogous to compress (stretch) the sequence $\mathbf{y}(t)$ along the time $t$ axis (Fig. 1B, left). $\alpha = 0$ is a particular case where the sequence remains at a constant value, analogous to "time freezing". $\alpha < 0$ implies "time reversal", i.e., flipping the sequence over the time axis and then scaling it based on the absolute value of $\alpha$ (Fig. 1B, right).

## 3 Disentangled neural representation of the sequence time and pattern

We focus on how temporal sequences $\mathbf{y}(t)$ with different temporal scales can be flexibly generated by canonical neural circuits. There are two aspects of temporal sequence generation: One is representing the temporal pattern (or content) of sequences referred to "pattern" representation, e.g., representing a song A vs. song B; and another is representing the time and time scale which is referred to as "time" representation, e.g., singing a song with normal speed vs. 2x speed. We reason the time and pattern of sequences can be represented separately in neural circuit [8], in that sequences can be scaled over time whatever their temporal patterns are. Therefore, we posit a *disentangled*, two-stage circuit architecture (Fig. 2A): A *recurrent* circuit acts as an "neural clock" whose population responses of $N$ neurons, $u(t) = \{u_j(t)\}_{j=1}^{N}$, represent the time information; and a *feedforward* circuit converts the spatiotemporal responses $u(t)$ into the sequence with the desired pattern, $\mathbf{y}(t)$, e.g., a concrete song or the trajectory of hand-written digits. Mathematically, the disentangled circuit can be denoted as,

$$\mathbf{y}(t) = F[u(t)], \tag{3}$$

where $F[\cdot]$ is the feedforward circuit storing the pattern information. Since $F[\cdot]$ is memoryless over time and maps the instantaneous response $u(t)$ into the instantaneous complex sequence value $\mathbf{y}(t)$, scaling the sequence $\mathbf{y}(\mathbb{t})$ over time can be realized by just scaling the recurrent circuit's response,

$$\mathbf{y}(\mathbb{t}) = \hat{S}(\alpha) \cdot \mathbf{y}(t) = \hat{S}(\alpha) \cdot F[u(t)] = F\Big[\hat{S}_u(\alpha) \cdot u(t)\Big], \tag{4}$$

where $\hat{S}_u(\alpha)$ is the TS operator acting on neural responses $u(t)$, in contrast with $\hat{S}(\alpha)$ acting on $\mathbf{y}(t)$.

In summary, realizing TS in the disentangled circuit only needs to modulate the recurrent circuit's response $u(t)$ without changing the feedforward circuit ($F[\cdot]$). In contrast, generating different patterns only needs to change (functional) feedforward circuits, while keeping the recurrent circuit's response $u(t)$ unchanged. The representation of time information in recurrent circuits was supported by a wide range of neurophysiology experiments in many brain areas, e.g., medial frontal cortex [6], hippocampus [24], striatum [25], and HVC in songbirds [26], etc. The disentangled architecture is also supported by a recent study on vocal motor circuits in a highly vocal rodent [27], where the motor cortex dictates the timing of songs, while the mid-brain or brain stem areas generate music notes (patterns).

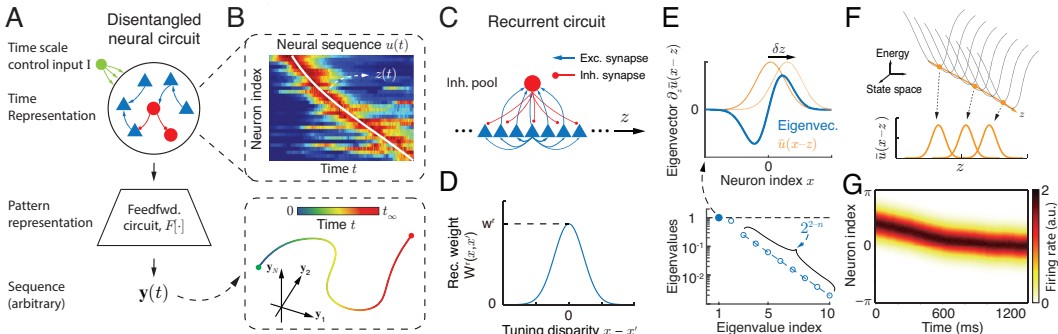

Figure 2: (A-B) A disentangled neural circuit that represents the sequence's time and pattern information in the recurrent and feedforward circuits respectively. The recurrent circuit generates a neural sequence embedding the "time" manifold $z$ (B top, adapted from [23]) regardless of the sequence pattern, and then the feedforward circuit maps the neural sequence into sequences with arbitrary patterns (B bottom). A control input (green circle) is supposed to modulate the temporal scale of the neural sequences generated by the recurrent circuit. (C) The recurrent circuit is modeled as a continuous attractor network (CAN) consisting of excitatory (E) and inhibitory (I) neurons. The E neurons are uniformly distributed in the one-dimensional $z$ manifold of the generic sequence (y-axis in top figure, panel B). (D) The recurrent connections E neurons, which decays with the difference between two neurons' preferred values of the generic sequence. (E) The eigenvalues and eigenvectors of the CAN, where the eigenvector with the largest eigenvalue corresponds to the movement of neuronal responses along the "time" manifold. (F) Energy landscape of the CAN without receiving the time-scale control input. (G) A generic neural sequence is generated by the recurrent circuit when receiving a time-invariant control input.

## 3.1 A low-dimensional "time" manifold in recurrent circuit's responses

We next investigate how the time is represented in $u(t)$ based on experimental observations. It was found that when generating movements with different time scales, recurrent circuit's responses $u(t)$ evolve along the same low-dimensional manifold with different speeds [2–7]. In contrast, the temporal responses of single neurons under different time scales may not align together. Therefore, we posit the one-dimensional (1D) time $t$ can be represented (embedded) in a 1D manifold in recurrent circuit responses $u(t)$. We denote by $z$ the 1D manifold of time representation in the circuit, in order to distinguish with the reference time $t$. It is worth noting that the neural response evolution on the $z$ manifold, $z(t)$, doesn't need to be linear with reference time $t$, and such a linear $z(t)$ also lacks experimental support. Mathematically, it only requires that $z(t)$ is a one-to-one (bijective) mapping with time $t$ [3]. Combined, the recurrent circuit's response $u(t)$ represents an internal "time" $z$, and in the rest of the paper, we will denote the response as a function of $z(t)$, i.e., $u[z(t)]$.

What is the internal "time" manifold $z(t)$ would look like? Experiments found there is a unique time for each neuron to reach its maximal response [2–7]. Sorting neurons based on the peak response timing, the population responses $u(t)$ resemble a bump-profile activity traveling across neurons (Fig. 2B, top). Hence, the 1D "time" manifold $z$ can be regarded as the axis of sorted neurons based on their peak timing (Fig. 2B, y axis). And the evolution of neurons' bump-profile responses on the $z$ manifold forms a sequence $z(t)$ representing the internal "elapsed time" [4, 5, 10, 14]. Eventually, the time can be explicitly read out by identifying the index of the neuron which reaches its maximal response, and the time scale ($\alpha$ in Eq. 1) is reflected by the speed of the neural sequence.

## 3.2 Temporal scaling operator acting on recurrent circuit's responses

As experiments suggested the TS transformations scale the "time" manifold $z$ rather than each individual neuron's temporal responses, we define $\hat{S}_u(\alpha)$ in Eq. (4) as,

$$u[z(\mathfrak{t})] = \hat{S}_u(\alpha) \cdot u[z(t)] = u[z(S(\alpha) \cdot t)] = u[z(\alpha t)]. \quad (5)$$

And such a neural representation $u[z(t)]$ is called *equivariant* with the temporal scaling group. To find the expression of $\hat{S}_u(\alpha)$, we consider an infinitesimal temporal scaling with $\alpha$ close to one ($\alpha \to 1$).

Reorganizing the scaled neural response as $u[z(\alpha t)] = u[z(\exp(\ln t + \ln \alpha))]$, and performing a first-order Taylor expansion with amount $\ln \alpha$,

$$u[z(\exp(\ln t + \ln \alpha))] \approx u[z(t)] + (\ln \alpha) \frac{du[z(t)]}{d \ln t} = \left[ 1 + (\ln \alpha) t \frac{dz}{dt} \frac{\partial}{\partial z} \right] \cdot u[z(t)]. \qquad (6)$$

From the above equation, the TS *generator* acting on neural responses can be defined as,

$$\hat{g}_u = t \frac{dz}{dt} \frac{\partial}{\partial z} \equiv t(\partial_t z)\partial_z, \qquad (7)$$

which characterizes the tangential direction of the TS effect on neural responses. Note that $\hat{g}_u$ cannot be simplified into $\hat{g}'_u = td/dt$ by canceling $dz$ and $\partial z$. This is because $\hat{g}_u$ only scales the 1D "time" manifold $z(t)$ in the high-dimensional neural response $u[z(t)]$, whereas $\hat{g}'_u$ scales every neuron's responses. TS operators with arbitrary $\alpha$ can be derived by composing many infinitesimal scaling transformations,

$$\hat{S}_u(\alpha) = \lim_{n \to \infty} \left[ \hat{S}_u(\alpha^{1/n}) \right]^n = \lim_{n \to \infty} \left( 1 + \frac{\ln \alpha}{n} \hat{g}_u \right)^n = \exp\left( \ln \alpha \cdot \hat{g}_u \right), \qquad (8)$$

Eq. (8) is also compatible with a negative $\alpha$ by using the generalized logarithmic function (see Eq. S9). To find the dynamics of the scaled neural responses, taking the time derivative of the operator,

$$\partial_t \hat{S}_u(\alpha) = \partial_t \exp\left( \ln \alpha \cdot \hat{g}_u \right) = \sum_n \frac{(\ln \alpha)^n}{n!} \partial_t \hat{g}_u^n = \sum_n \frac{(\ln \alpha)^n}{n!} (1 + \hat{g}_u)^n (\partial_t z)\partial_z,$$

$$= \exp[(\ln \alpha)(1 + \hat{g}_u)](\partial_t z)\partial_z = \alpha \hat{S}_u(\alpha)(\partial_t z)\partial_z, \qquad (9)$$

where we utilize the result that $\partial_t \hat{g}_u^n = (1 + \hat{g}_u)^n (\partial_t z)\partial_z$ (see Eq. S13 in SI). And then the dynamics of the scaled equivariant neural sequence is,

$$\partial_t u[z(\mathbb{t})] = \partial_t \hat{S}_u(\alpha) \cdot u[z(t)] = \alpha \hat{S}_u(\alpha)(\partial_t z)\partial_z \cdot u[z(t)] = \alpha[\partial_{\mathbb{t}} z(\mathbb{t})]\partial_{\mathbb{z}} \cdot u[z(\mathbb{t})]. \qquad (10)$$

Scaling the neural dynamics will modulate its time derivative by $\alpha$, and the right-hand side (RHS) of the scaled dynamics (Eq. 10) depends on the scaled $z(\mathbb{t})$ rather than the original $z(t)$. Although the TS effect on the dynamics can be also derived via the chain rule (e.g., [20]), deriving such an effect from Lie group give us more insight on the algebraic structure of TS and its circuit representation.

### 3.3 A concrete neural representation of internal "time"

We next define a concrete neural representation of internal "time" manifold $z$ based on experimental observations. We consider a parametric representation where each neuron is selective for the internal $z$ value, and denote $x_j$ as the preferred $z$ value of the $j$-th neuron which implies the neuron will reach its peak firing rate when $z(t) = x_j$. Moreover, we consider the preferences of neurons, $\{x_j\}_{j=1}^N$, are *uniformly* distributed along the $z$ manifold to simplify the math analysis. Since the population response has a bump-profile (Fig. 2B, top), we model the mean firing rate (tuning) of $j$-th neuron as a Gaussian function over $z(t)$, a widely used approach in neural coding studies (e.g., [28–30]),

$$\bar{u}_j[z(t)] = A_u \exp[-(x_j - z(t))^2 / 4a^2], \qquad (11)$$

where $A_u$ is the peak firing rate, and $a$ the tuning width. Note that $\bar{u}_j$ regards as the mean neuronal response, in distinguish with the instantaneous $u_j(t)$ which may deviate from the mean response. Since Lie group operators act on continuous functions (Eq. 5), to simplify the math analysis, we treat $\bar{u}[z(t)]$ as a continuous function corresponding to an infinite number of neurons in the population ($N \to \infty$), and all neurons' preferences become a continuous variable, i.e, $x_j \to x$. Hence, the mean population firing rate of all neurons becomes a continuous function of the $z(t)$,

$$\{\bar{u}_j[z(t)]\}_{j=1}^N \equiv \bar{u}[z(t)] = A_u \exp[-(x - z(t))^2 / 4a^2]. \qquad (12)$$

## 4 A temporal scaling equivariant recurrent neural circuit model

We propose a biologically plausible recurrent circuit dynamics with *fixed* connectivity to flexibly generate TS equivariant neural sequences $u[z(t)]$, and study how the circuit can be controlled to scale the neural sequence. Specifically, the requirements specified by the TS group and neural representations (Eqs. 5- 12) will be used as objectives of the proposed functional recurrent circuit.

## 4.1 The recurrent circuit dynamics

First, we consider a special case of freezing the internal "time" by applying $S(0)$, and hence $z(t) = z(t_0)$ as the initial value. This can be regarded as, e.g., the arm position is fixed over time. Then the mean, stationary neuronal responses in the recurrent circuit given the $z(t_0)$ need to match the Gaussian profile as indicated by Eq. (12). Moreover, since the initial value $z(t_0)$ is arbitrary (e.g., the arm location can be stable at any location at any time), stationary neural responses are required to be stable at any value of $z$. This implies that the recurrent circuit dynamics should have a family of stationary states (attractors) along the $z$ manifold. These kinds of networks are called continuous attractor networks (CANs), which have been widely used to explain the neural coding of continuous features (e.g., [31–33]). We firstly consider an autonomous CAN dynamics as follows [34–37],

$$\tau \partial_t u(x,t) = -u(x,t) + \rho \int J(x,x')r(x',t)dx', \tag{13a}$$

$$r(x,t) = G[u(x,t)] = \frac{[u(x,t)]_+^2}{1 + k\rho \int [u(x',t)]_+^2 dx'}. \tag{13b}$$

where $u(x,t)$ is the synaptic input received by the excitatory neuron preferring $x$ (Fig. 2C, blue triangle), and $\tau$ is the time constant. $G(\cdot)$ is a nonlinear activation function modeled as divisive normalization, a canonical operation in the cortex [38], where $k$ is the global inhibition strength coming from an inhibitory neuron pool (Fig. 2C, red disk). $[\cdot]_+^2$ denotes rectifying the negative part followed by a square function. $\rho$ is the neuronal density on the $z$ manifold. $J(x,x')$ denotes the synaptic weight from the pre-synaptic neuron preferring $x'$ to the post-synaptic neuron preferring $x$, and is modeled as a Gaussian function over the difference $|x - x'|$ (Fig. 2D),

$$J(x,x') = J(x - x') = J_0 \exp[-(x - x')^2/2a^2]. \tag{14}$$

The $J_0$ and $a$ in Eq. (14) denote the peak recurrent weight and the connection width respectively. It can be checked that the autonomous recurrent circuit can self-consistently generate non-zero, desired Gaussian profile population responses $\bar{u}(z)$ as specified in Eq. (12),

$$\bar{u}(x - z) = A_u \exp[-(x - z))^2/4a^2] = \bar{u}(z), \tag{15}$$

as long as the peak recurrent weight $J_0$ is larger than a critical value $J_c$ (can be analytically solved, see Eq. S30 for details). Note that the $z$ in the above equation is a free parameter, meaning the recurrent circuit model has a continuous family of attractors over the "time" $z$ manifold (Fig. 2F). Here we use the notation $\bar{u}(x - z)$ to denote the responses generated by the concrete recurrent circuit dynamics (Eqs. 13a-13b), in distinguishing with $\bar{u}[z(t)]$ (Eq. 12) directly specified in earlier derivations.

To generate a moving neural sequence (dynamic $z(t)$; Fig. 2B, top), one possibility is applying a *time-invariant* control input $I(x)$ to the recurrent circuit (Fig. 2A, green disk), i.e., inserting $I(x)$ into the RHS of Eq. (13a) (see Discussion for other mechanisms of generating moving neural sequences),

$$I(x) = I(x|z_\infty) = \mathrm{I}_0 \exp[-(x - z_\infty)^2/4a^2]. \tag{16}$$

$\mathrm{I}_0$ is the peak controlling input intensity, and $z_\infty$ is the final position of the neural sequence (the final location at y-axis of Fig. 2B, top). Then if the neural response is initiated at $u(x,t_0) = \bar{u}[x - z(t_0)]$, it will move along $z$ manifold towards $z_\infty$ and generates a neural sequence. Fig. 2G shows a moving neural sequence generated by the recurrent circuit model when receiving such a control input.

## 4.2 Identify temporal scaling operators in the recurrent circuit dynamics

We pursue a theoretical understanding of how the recurrent circuit dynamics represents temporal scaling operators. Since an TS operator $\hat{S}_u(\alpha)$ is an exponential map based on the TS factor $\alpha$ and the TS generator $\hat{g}_u$ (Eq. 8), we specifically identify how the recurrent circuit represents $\alpha$ and $\hat{g}_u$.

The TS generator (Eq. 10) requires the neural dynamics only preserves perturbations along the $z$ manifold while filters out perturbations along other directions. This is because the time derivative of the neural response is only proportional to the partial derivative over the 1D $z$ manifold, i.e., setting $\alpha = 1$ in Eq. (10) yields $\partial_t u[z(t)] = (\partial_t z)\partial_z u[z(t)]$. To test whether this requirement can be satisfied by the proposed recurrent circuit, we performed perturbative analysis of the circuit dynamics by considering the instantaneous network state as perturbed from the attractor state, i.e.,

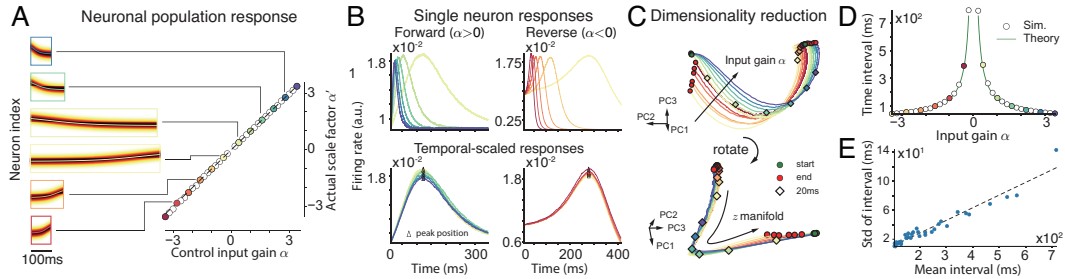

Figure 3: Temporal scaling of the neural sequence in the proposed recurrent circuit by gain modulation on the time-invariant control input. (A) Left: neural sequence generated with different control input gains. Right: The actual temporal scale factor with the control input gain. (B) An example excitatory neuron's temporal responses (top) and responses scaled over time (bottom) in forward (left) and reverse (right) conditions. Color: input gain as suggested by dots in (A). (C) The dimensionality reduction of the neuronal responses at different control input gains. (D) The time interval (absolute value) decreases with control input gain. (E) Weber's law in the recurrent circuit where the standard deviation of the time interval is proportional to the mean interval.

$u(x, t) = \bar{u}(x - z) + \delta u(x, t)$ (ignoring the time dependence on $z$ for brevity). Substituting this expression into the network dynamics (Eq.13a) yields the perturbation dynamics,

$$\tau \partial_t \delta u(x, t) = -\delta u(x, t) + \int K(x, x'|z) \delta u(x', t) dx', \qquad (17)$$

where $K(x, x'|z) = \rho \int J(x - x'') \partial \bar{r}(x'' - z) / \partial \bar{u}(x' - z) dx''$. Treating $K(x, x'|z)$ as an operator acting on $\delta u(x, t)$, its eigenvalues $\lambda_n$ can be calculated as (descending order, Fig. 2E) [37, 39],

$$\lambda_1 = 1, \quad \lambda_2 = 1 - \sqrt{1 - J_c^2/J_0^2}, \quad \lambda_n = 2^{2-n} \ (n \geq 3), \qquad (18)$$

and the (unnormalized) eigenfunction $f_1(x|z)$ with the largest eigenvalue $\lambda_1 = 1$ is calculated as,

$$f_1(x|z) \propto \partial_z \bar{u}(x - z) \propto -(x - z) \exp[-(x - z)^2/4a^2], \qquad (19)$$

which is the partial derivative of neural response over the $z$ manifold. The above analysis proves the CAN does satisfy the requirement of the TS generator approximately (Eq. 8), i.e., only preserving the perturbation along the $z$ manifold ($\lambda_1 = 1$) and removing other perturbations since their eigenvalues are smaller than 1. This is a unique property of the CAN which comes from the translation-invariant recurrent weights along the $z$ manifold (Eq. 14).

We present simple derivations to identify the TS generator in the recurrent circuit with a control input (rigorous derivations, SI. Sec. 3). The control input will deviate the network state $u(x, t)$ from the attractor state $\bar{u}(x - z)$. Hence, our theory considers the weak limit of control input (small $I_0$), and substitute the attractor states (Eq. 15) into the circuit dynamics (Eq. 13a),

$$\tau \partial_t \bar{u}(x - z) \approx [-\bar{u}(x - z) + \rho J * \bar{r}(x - z)] + I(x|z_\infty) \approx I(x|z_\infty), \qquad (20)$$

where the decaying term $-\bar{u}(x - z)$ and the recurrent input term $\rho J * \bar{u}(x - z) = \rho \int J(x, x') \bar{r}(x' - z) dx'$ will cancel each other due to the definition of attractor states (Eq. 15). Then we treat the control input $I(x|z_\infty)$ as a "perturbation" of the attractor state, and decompose it using eigenfunctions,

$$\tau \partial_t \bar{u}(x - z) \approx I(x|z_\infty) = \sum_n a_n f_n(x|z) \approx a_1 f_1(x|z) = a_1 \partial_z \bar{u}(x - z), \qquad (21)$$

where we ignore perturbations along directions parallel to eigenfunctions $f_n(x|z)$ ($n \geq 2$) because they will eventually vanish (Eq. 18). The coefficient $a_1$ can be obtained by computing the inner product between the left and right-hand side of the above dynamics with $f_1(x|s)$, e.g., $a_1 = \langle \tau \partial_t \bar{u}(x - z), f_1(x|z) \rangle = \tau \int \partial_t \bar{u}(x - z) f_1(x|z) dx$,

$$a_1 = \tau \partial_t z = I_0 \cdot A_u^{-1}(z_\infty - z) \exp[-(z_\infty - z)^2/8a^2]. \qquad (22)$$

Combining the above two equations, the proposed recurrent circuit with the control input approximately achieves the TS equivariance: its attractor state $\bar{u}(x - z)$ (Eq. 15) travels along the "time" $z$ manifold in a way consistent with the scaled dynamics derived from TS group (Eq. 10, $\alpha = 1$),

$$\partial_t \bar{u}(x - z) \approx (\partial_t z) \partial_z \bar{u}(x - z).$$

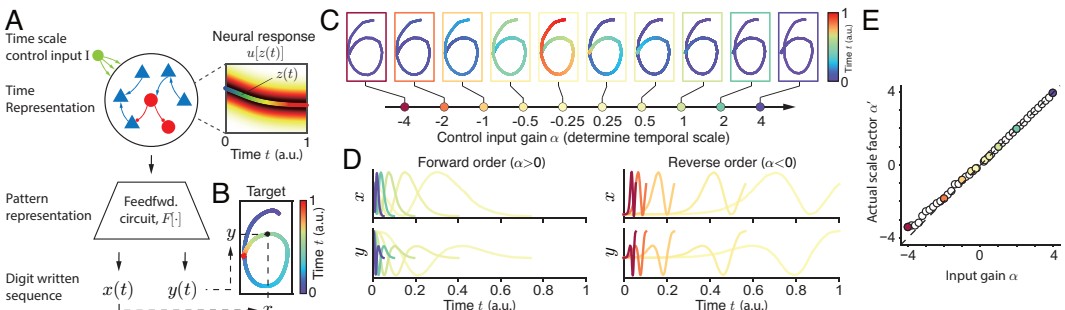

Figure 4: Generating a sequence pattern by feedforward circuit mapping. (A-B) A proof of concept example where the feedforward circuit transforms the neural sequence in the recurrent circuit into the $x$ and $y$ coordinates of a hand-written sequence of the digit '6'. (C) The hand-written sequence at different control input gains in the recurrent circuit. Color: elapsed time $t$. (D) The temporal trajectories of the hand-written sequence at forward (left) and reverser (right) order. (E) The actual temporal scale factor of hand-written sequence is proportional to the control input gain.

In practice, the actual responses $u(x, t)$ will deviate from the attractor state $\bar{u}(x - z)$ due to the control input and noises. The recurrent dynamics will quickly remove distortions perpendicular to the $z$ manifold, corresponding to pull $u(x, t)$ back to the continuous $z$ manifold, and then the attractor state $\bar{u}(x - z)$ will travel along the $z$ manifold which internally represents "elapsed time".

**The emergence of the TS generator**. A non-vanishing TS generator $\hat{g}_u = t(\partial_t z)\partial_z \propto t a_1 \partial_z$ (Eq. 8) needs a non-zero $a_1$, which can be satisfied with a non-zero offset, $z_\infty - z$, between the final state $z_\infty$ and the initial state $z_0$. Therefore, the spatial offset of control input and the initial network state on the $z$ manifold emerges the TS generator in the recurrent circuit dynamics.

**The representation of the TS factor**. Scaling neural sequences over time corresponds to multiplying the internal "time" dynamics $\partial_t z \propto a_1$ by the scaling factor $\alpha$ (Eq. 10). Since $a_1$ is proportional to the control input strength $I_0$ (Eq. 22), scaling the neural sequence can be realized by the *gain modulation* on the control input by the scaling factor $\alpha$, i.e., $I(x|z_\infty) \to \alpha I(x|z_\infty)$ (Eq. 16). And then the whole TS equivariant recurrent circuit dynamics becomes ($*$ denotes the convolution over $x$),

$$\tau \partial_t u(x, t) = -u(x, t) + \rho J * r(x, t) + \alpha \cdot I_0 \exp[-(x - z_\infty)^2 / 4a^2]. \tag{23}$$

Therefore, the TS factor $\alpha$ determines the control input gain in the recurrent circuit. Generating sequences with different temporal scales can be realized by simple gain modulation, which is a widely observed operation in neural circuits (e.g., [40, 41]). Interestingly, by applying a negative input gain ($\alpha < 0$) the circuit dynamics can implement "time reversal", i.e., a flipped sequence over time.

## 5 Simulation experiments

We simulated the proposed circuit model to test temporal scaling via manipulating the control input's gain (SI. Sec. 4.1, network simulation detail). Changing the input gain $\alpha$ varies neural sequences' time scales (Fig. 3A, left), and the actual scaling factor is proportional to the input gain $\alpha$ as predicted by our theory (Fig. 3A, right). It is noteworthy that a negative gain enables the circuit to render a "time reversed" neural sequence, corresponding to flipping the original sequence on the time axis (comparing heatmaps in Fig. 3A, left). At the single neuron level, temporal scaling also changes the temporal speed of single neurons' responses (Fig. 3B). The single neuron's temporal responses under gains with the same polarity, but not different polarities, can be scaled and overlap well over time. Moreover, the dimensionality reduction analysis shows neural sequences under both positive and negative control input gain all evolve along the $z$ manifold and overlap perfectly (Fig. 3C). These results are consistent with the experimental finding that single neurons responses might not overlap well under different time scales, but the population responses at different time scales all evolve along the same low-dimensional manifold and overlap perfectly [2–7].

A characteristic of temporal scaling at the behavioral level is Weber's law, meaning the standard deviation of the time duration of a sequence linearly increases with the mean duration. To reproduce this phenomenon in the model, we include independent multiplicative noise in the recurrent circuit

mimicking the Poisson spike variability (inserting $\sqrt{\tau \mathsf{F}[u(x,t)]_+}\xi(x,t)$ to the RHS of Eq. (23) with $\mathsf{F}$ the Fano factor). The mean time duration (Eq. S34, analytical solution) decreases with the input gain $\alpha$, implying the internal "time" representation becomes faster (Fig. 3D). Meanwhile, the standard deviation of the time duration is proportional to the mean duration, consistent with Weber's law.

The recurrent circuit's population responses $u[z(t)]$ can be mapped to an arbitrary, complex sequence $\mathbf{y}(t)$ through a feedforward circuit (Eq. 3). As a proof of concept, we use a feedforward circuit (modeled as a three-layer perceptron) to transform $u[z(t)]$ into a 2D sequence ($x$ and $y$ coordinates) of hand-written digits, e.g., digit "6" (Fig. 4A-B, see details in SI. Sec. 4.3). The feedforward circuit was trained via back-propagation by only using the neural response and the hand-written sequence (Fig. 4B) at only one temporal scale. After training, we test whether the whole disentangled circuit can generalize the hand-written "6" sequence at other time scales by manipulating the control input's gain ($\alpha$ in Eq. 23). Indeed, the feedforward circuit trained at one temporal scale successfully generates hand-written sequences over other time scales (Fig. 4C-D), including both positive (forward sequence) and negative (reversed sequence) temporal scales. The disentangled circuit can also generate hand-written sequences of other digits once we retrain the feedforward circuit (see SI. Figures).

## 6  Conclusion and Discussion

The present study investigates the neural circuit mechanism of temporal scaling group equivariant representation. We propose a disentangled neural circuit that represents the pattern (content) and the time information of temporal sequences separately. The timing information is equivariantly represented by neural sequences generated in a recurrent circuit modeled as a CAN, and the sequence pattern (content) is represented in a feedforward circuit that maps the generic neural sequence to arbitrary temporal sequences. Moreover, TS operators are explicitly represented as a time-invariant control input applied to the CAN: the TS factor determines the control input gain, and the TS generator emerges by the spatial offset between the control input and network state along the continuous attractor manifold representing "time". Eventually, modulating the control input gain enables the recurrent circuit with fixed connectivity to generate sequences with different temporal scales, including "time reversal" when using a negative gain. Our study for the first time formulates the temporal scaling problem from the Lie group, and links the abstract group to a concrete, biologically plausible neural circuit. It gives us insight into the math principle underlying TS representation in neural circuits, and the proposed model may inspire a new building block for equivariant representation in ML tasks.

**Comparison to other works.** How temporal scaling is achieved in neural circuits has been an active topic (e.g., [2, 3, 6, 9, 13, 14, 42–44]. A large body of earlier neural circuit modeling studies investigated this question via training a recurrent network, e.g., a reservoir network, to represent different time scales (e.g., [6, 9, 11, 14]). Although the trained networks achieve the TS representation, the underlying math principle is still lacking. In contrast, the present paper *analytically* derives a recurrent circuit implementation of TS equivariant representation, and the explicit representation of TS group operators. Moreover, our hand-crafted circuit model generalizes over temporal scales well, which outperforms recent neural circuit models where the trained network cannot generalize (extrapolate) well to temporal scales beyond the range in the training set [11, 27]. In addition, previous studies considered generating a moving neural sequence by adding an anti-symmetric recurrent weight component in the recurrent network (e.g., [45–48]), whose magnitude increases with temporal speed (or TS factor) [37, 47]. Nonetheless, the TS in the brain happens in short time scales (milliseconds to a second), which is too short to modulate recurrent weights. In contrast, the control input gain in the present study can be quickly modulated in short time scales. Last, previous studies proposed TS covariant networks, inspired by the fact that the width of the temporal field of hippocampal time cells covaries (increases) with the preferred elapsed time (e.g., [22]). In comparison, the TS covariant networks aim to achieve TS invariant representation that is beneficial for recognizing sequences with different temporal scales, while our network model is flexibly generating sequences with different temporal scales. When inverting the information flow and introducing a feedback loop between the control input and the recurrent network, our model also has the potential to recognize (infer) the input sequences and their temporal scaling factors.

Equivariant representation is an active topic in ML research and lots of equivariant neural networks have been proposed (e.g., [49–53]). Nonetheless, to our best knowledge, previous ML studies exclusively investigated the equivariance under group transformations in spatial domain rather than

temporal domain. Moreover, there have not been many ML studies investigating recurrent network mechanisms of equivariance while most of them studied feedforward equivariant networks. Timing or TS representation in ML models typically uses a mechanism called time warping, which relies on a pacemaker (central clock) producing sine waves with different frequencies to represent times. However, the pacemaker mechanism doesn't exist in the nervous system representing time scales ranging from the order of milliseconds to seconds [3]. There are recent ML studies investigated temporal scaling recurrent networks [20], where they directly modulated the time constant of neural dynamics (i.e., $\tau$ in Eq. 13a). However, the time constant of real neurons hardly changes due to biophysical properties. And our way of achieving TS by gain modulation of the amplitude of the time-invariant control input is novel.

**Limitations of the model.** In order to gain a theoretical understanding of the TS equivariant representation, we propose a hand-crafted recurrent circuit model rather than training a neural network model as in previous studies [9–11, 14, 48]. Moreover, we consider a simple model whose recurrent connections do not have random components (Eq. 14). It is interesting to study how a TS equivariant recurrent network can be trained in an unsupervised way. In addition, previously trained network models used reservoir networks containing strong random connections with disordered (chaotic) dynamics [9–11, 14, 48]. Although chaos provides a rich dynamical repertoire to facilitate learning, it impairs the robustness of time representation [9]. Since this study does not consider learning, ignoring the random connections will significantly help us identify how structured recurrent connections achieve TS equivariance and represent operators. Statistically, a characteristic of chaotic spiking networks is the Poisson spiking variability [54, 55], whose statistical effect is equivalent to the injected multiplicative noise in our model. Furthermore, as a proof of concept, the feedforward circuit used in our model is simple and needs to be retrained to generate a new sequence pattern, e.g., other digits' written sequence, which is unlikely in the brain. One way to improve this is by using a compositional model (e.g., [56, 57]) which can be flexibly modulated to change its represented pattern.

**Extensions of the model.** The proposed neural circuit model has the potential to explain other brain functions. For example, our model may be used to explain the memory formation in the hippocampus, where it is believed that hippocampal theta oscillation compresses (speeds up) sensory input sequences by about 10 times and is beneficial for memory formation [58]. It is possible that the theta oscillation shares a similar computational mechanism with the proposed temporal scaling equivariant recurrent network. At last, experiments found "time" cells in the hippocampus whose temporal firing width (duration) is proportional to their peak timing, i.e., a "time" cell fires later will fire longer, and time cells' preferred peak timing is log-scaled [59]. Such a relation doesn't exist in the proposed model, and neurons in the current model can be "deployed" in the logarithmic space of $z$ manifold to reproduce this effect. All of these will form our future research.

# Acknowledgments

Y. W. is supported by NSF DMS-2015577. S. W is supported by National Science and Technology Innovation 2030 Major Program (No. 2021ZD0200204, Si Wu).

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
