# OpenReview forum: "A Recurrent Neural Circuit Mechanism of Temporal-scaling Equivariant Representation"
_NeurIPS.cc/2023/Conference — NeurIPS 2023 poster_

### Official Review · Reviewer_NfgN · 2023-06-27

**Soundness:** 4 excellent
**Presentation:** 3 good
**Contribution:** 3 good
**Rating:** 7
**Confidence:** 2

**Summary:**

Empirical data have shown the existence of neurons that represent time and the ability of neural circuits to generate temporal sequences at different speeds. The authors propose a model that takes into account these findings: a continuous attractor network that acts as a neural clock which can evolve with a speed proportional to its input, and a feedforward neural network that use this signal to generate a pattern. In particular, they propose a continuous attractor architecture that has attracting biologically plausible properties, analyze it from a Lie group perspective and show in simulation that their theoretical arguments hold in practice.

**Strengths:**

The proposed approach makes a lot of intuitive sense (I am actually surprised that this approach has not been proposed before) and match experimental data. The analysis is comprehensive and illustrated with well designed figures that help getting the gist of it.

**Weaknesses:**

What does the Lie group perspective bring? To the best of my understanding, the Lie group perspective is not used in the analysis and requires introducing additional jargon that makes the paper harder to read. Can the authors clarify the interest of the TS group?

How novel is the idea that input gain modulates moving speed of the bump? From my understanding as an outsider of the field, similar ideas have been introduced to model head direction cells. The authors do not seem to include such references. Is it a misunderstanding from my side or are some references missing?

I’m happy to increase my score once those questions are answered.

**Questions:**

--

**Limitations:**

Yes

---

> ### Author Rebuttal · Authors · 2023-08-08
>
> We are grateful to your positive review and honored to answer your questions.
>
> > “What does the Lie group perspective bring? To the best of my understanding, the Lie group perspective is not used in the analysis and requires introducing additional jargon that makes the paper harder to read. Can the authors clarify the interest of the TS group? ”
>
> Thank you for highlighting this aspect of our research. While we indeed propose a recurrent network for the TS transformation, the true essence and significance of our study lie elsewhere. We delve deeper to furnish a **theoretical** and **analytical** framework, elucidating the TS group representation within recurrent network dynamics (lines 231 – 243).
> The Lie group's role is paramount here. It's not just an ancillary component but the theoretical basis that enables us to derive the TS operations on neural representations, as illustrated by Eqs. 7-10. By comparing this with our theoretical analysis of network dynamics (as per Eq. 23), we're able to pinpoint the TS group operator representation. Such insights, which bridge abstract group representation with concrete, biologically plausible recurrent network dynamics, immensely further our grasp of the **algebraic structure** within these dynamics. Furthermore, this understanding will be pivotal as we investigate the neural circuit representations with even more intricate group structures in future research.
> We'd like to underline that, to our knowledge, our study pioneers this unique theoretical insight into TS group representation—filling a significant gap in the existing literature.
>
> > “How novel is the idea that input gain modulates moving speed of the bump? From my understanding as an outsider of the field, similar ideas have been introduced to model head direction cells. The authors do not seem to include such references. Is it a misunderstanding from my side or are some references missing?”
>
> While prior neuroscience research has employed gain modulation for temporal scaling within recurrent network dynamics—as acknowledged in our references (Ref. 6, 9, 10, 13)—our study introduces two distinct and novel contributions:
> 1. **Time Reversal through Negative Gain**: Our findings indicate that a negative gain can effectuate a "time reversal." To the best of our knowledge, this phenomenon hasn't been delineated in preceding studies.
> 1. **Analytical Representation of Temporal Scaling Factor**: We rigorously derive that the input gain symbolizes the temporal scaling factor, denoted as $\alpha$ associated with TS operators. While this might be conceptually intuitive to some, our study offers the first mathematically rigorous representation of this relationship.

---

> > ### Comment · Reviewer_NfgN · 2023-08-12
> >
> > Thanks to the authors for answering my questions.
> > I increase my score to 7.

---

> > > ### Author Response · Authors · 2023-08-12
> > >
> > > We're truly thankful for your revised score and positive feedback on our paper!

---

### Official Review · Reviewer_aiKz · 2023-07-06

**Soundness:** 3 good
**Presentation:** 3 good
**Contribution:** 2 fair
**Rating:** 4
**Confidence:** 2

**Summary:**

This paper introduces a continuous attractor neural network that has a simple way of generating a specific sequence of outputs at a speed tunable by changing the gain of a constant (but neuron-dependent) input. The authors show how this can be used in conjunction with a multi-layer perceptron to generate arbitrary sequence at speeds that can be tuned by controlling the gain. A negative gain can be used to reverse the sequences. The authors analyze their model from the perspective of the Lie group of temporal scaling.

**Strengths:**

*Originality:*
The paper introduces a novel way of controlling the speed at which a network generates a temporal sequence by separating it into a component that is equivariant with time scaling and one that is independent of time, and simply transforms the output from the first component.

*Quality:*
The paper uses detailed mathematical derivations as well as simulations to support its claims.

*Clarity:*
The presentation is clear, with detailed proofs and derivations in the Appendix. The code for the simulations is also included in the Supplementary Material (which should be standard but sadly is not true for all submissions).

*Significance:*
The work sheds more light on how brains can generate sequences at tunable speeds, which is important in the study of behaviors from song to speech.

**Weaknesses:**

I find that the machinery of Lie group theory is not really needed for this work and potentially makes it harder to understand. As I understand it, the point is that a continuous attractor network with a normalizing nonlinearity can be used to generate sequences with tunable speeds by simply controlling the gain of a specially tailored input. The need for Lie group theory was not very clear, beyond perhaps the simple point of using the infinitesimal version of a scaling transformation.

**Questions:**

Once in the text (below eq. (21)) and then again in the Appendix, the claim is made that the eigenvectors of the operator $K(x,x'\vert z)$ from eq. (17) are orthogonal. This does not seem obvious (or even possible?), since the operator is, as far as I can tell, not symmetric. Am I missing something?

Other comments:
* in Figure 1B, I suggest putting the original on top
* rotating panel A of Figure 1 by 90º could also help so that it aligns with panel B
* in line 102: reference [23] is not about birds, but rodents
* the notation $[u(x,t)]_+^2$ is used – I'm assuming the '+' stands for rectification, but this seems superfluous since the quantity is squared anyway; if the rectification is indeed not necessary, I suggest taking the '+' sign out to avoid confusion
* caption of Figure 4: "The temporal trajectories of the hand-written sequence at forward *(top)* and reverse *(bottom)* order" -- should be *(left)* and *(right)*
* on lines 210–211: it seems like the authors are referring to an approximation where only the component along the top eigenvector is kept; if so, this should be made clear

**Limitations:**

The authors have adequately discussed limitations of their work.

---

> ### Author Rebuttal · Authors · 2023-08-08
>
> Thanks for your positive comments about the strengths, especially the analytical derivations, of our work. We will revise the manuscript accordingly based on your suggestions. Below please finds our detailed reply for your comments.
>
> > "I find that the machinery of Lie group theory is not really needed for this work and potentially makes it harder to understand. As I understand it, the point is that a continuous attractor network with a normalizing nonlinearity can be used to generate sequences with tunable speeds by simply controlling the gain of a specially tailored input. The need for Lie group theory was not very clear, beyond perhaps the simple point of using the infinitesimal version of a scaling transformation."
>
> We appreciate the reviewer's summary of our work from a network implementation perspective. Admittedly, resorting to Lie group theory might seem excessive when the objective is perceived as merely formulating a network model for temporal scaling (TS) transformations. However, the true depth and ambition of our study span a broader horizon.
> 1. **Introduction of Equivariance**: Our study stands at the vanguard by rigorously introducing equivariance to delineate TS. Recently, the concept of equivariance has garnered substantial attention in the ML community (Cohen & Welling, 2016). Its applications in geometric machine learning, molecular design (Hoogeboom, 2022), fluid dynamics (Wang, 2020), and particle physics (Bogatskiy, 2020) have already been established. Consequently, leveraging Lie group theory to demystify cognitive computations promises to catalyze foundational advancements in neuroscience research.
> 2. **Continuous Attractor Dynamics via Lie Group Framework**: Our research meticulously derives the continuous attractor dynamics through the prism of the Lie group framework, offering an innovative approach to TS transformations. This novelty has also been recognized and endorsed by Reviewer ZBA5.
> 3. **Theoretical and Analytical Insights**: The cornerstone of our research lies in providing a robust **theoretical** and **analytical** understanding of the TS group representation within recurrent network dynamics (lines 231 – 243). It's pivotal to note that the Lie group is essential to this exploration. Through the Lie group, we distilled the TS operations on neural representations (Eqs. 7-10) and juxtaposed them with the theoretical analysis of network dynamics (Eq. 23), leading us to discern the TS group operator representation. Such a theoretical discourse on the abstract group's representation in concrete, biologically plausible network dynamics paves the way for a more profound comprehension of the **algebraic** structure of network dynamics. This perspective is instrumental when aiming to construct or understand network representations of intricate group structures in the future. To the best of our knowledge, no extant research has ventured into or achieved such profound theoretical insights into TS group representation.
>
> In summary, while our approach might seem intricate on the surface, the depth and breadth of its insights, and the gaps it fills in the current literature affirm the significance of our methodology.
>
> > "Once in the text (below eq. (21)) and then again in the Appendix, the claim is made that the eigenvectors of the operator $K(x,x'|z)$ from eq. (17) are orthogonal. This does not seem obvious (or even possible?), since the operator is, as far as I can tell, not symmetric. Am I missing something?"
>
> Reply: Thank you for pointing this out. The eigenfunctions are not orthogonal since the operator $K(x,x’|z)$ is not symmetric, but this does not affect our theoretical result in Eq 22. In the theoretical analysis, we used an orthonomal basis function set $v_n(x|z)$ (Eq. S22) to find the eigenfunctions $f_n(x|z)$ of $K(x,x’|z)$, and then the eigenfunctions are expressed as linear combinations of $v_n(x|z)$. Importantly, the eigenfunction of the "time" manifold, $f_1(x|z)$, is parallel to the basis function $v_1(x|z)$. Therfore the calculation of $a_1 = \langle \tau \partial_t \bar{u}(x-z), f_1(x|z) \rangle$ utilizes the **orthogonal basis function set**, rather than orthogonal eigenfunctions.
>
> We will correct the typo by removing the sentence "since eigenfunctions $f_n(x|z)$ are orthogonal to each other" in line 221.
>
> **Reply to comments on typos, figures and notations:**
>
> > “in Figure 1B, I suggest putting the original on top”
> > “rotating panel A of Figure 1 by 90º could also help so that it aligns with panel B”
>
> Thanks for the suggestions. We will revise our manuscript accordingly.
>
> > "in line 102: reference [23] is not about birds, but rodents”
>
> Thanks for your advice.  We will revise it accordingly.
>
> > “the notation is used $[u(x,t)]_+^2$ … to avoid confusion”
>
> Yes, the index $+$ denotes the rectification. It is necessary to rectify the negative $u(x,t)$ before taking the square. For example, say $u(x,t)=-1$, and $[u(x,t)]_+^2 = 0$ with rectification before square, whereas $[u(x,t)]^2 = 1$ will yield a non-zero positive output.
>
> > “caption of Figure 4: … -- should be (left) and (right)”
>
> Thanks for pointing that out.  We will revise it accordingly.
>
> > “on lines 210–211: it seems like the authors are referring to an approximation where only the component along the top eigenvector is kept; if so, this should be made clear”
>
> Thanks for the suggestion and we will revise the text to make it clearer. We mentioned this in line 220 when we only consider the component along the 1st eigenvector. To emphasize this approximation, we plan to revise the text in lines 109-111 as: \
> _"The above analysis proves the CAN does satisfy the requirement of the TS generator approximately (Eq. 8), i.e., only preserving the perturbation along the $z$ manifold $(\lambda_1=1)$ and removing other perturbations since their eigenvalues are smaller than $1$."_

---

> > ### Comment · Reviewer_aiKz · 2023-08-11
> >
> > Thanks for your explanations! Regarding the notation $[u(x,t)]^2_+$: I understand now, the rectification is applied before the squaring. That, however, is not obvious from the notation. I suggest either $u_+(x,t)^2$ or more explicitly $\text{relu}(u(x,t))^2$.

---

> > > ### Author Response · Authors · 2023-08-11
> > >
> > > Thanks for your suggestion. We will revise it into $ReLu[u(x,t)]^2$ to make it clearer.

---

> > > ### Author Response · Authors · 2023-08-18
> > >
> > > Dear Reviewer aiKz,
> > >
> > > Thanks for reviewing our paper!
> > >
> > > We are wondering whether our reply addresses your major concern, i.e., the necessity of using the Lie group in this study.
> > > One contribution of our study is to analytically identify how temporal scaling operators can be represented in biologically plausible recurrent neural circuit dynamics, which, to our best knowledge, has not been investigated in the previous study. We believe having such a theoretical result gains our understanding of the algebraic structure of neural circuits, and inspires us to study circuit implementation of more complicated Lie group operators in the future.
> > >
> > > If you have other concerns, feel free to bring them up and we are happy to explain.
> > >
> > > Thanks again!
> > >
> > > Authors of paper #3140

---

> > > > ### Comment · Reviewer_aiKz · 2023-08-18
> > > >
> > > > I understand the potential utility of Lie group theory in the future, but it still seems rather overpowered for the current study. Thanks for your detailed explanations, but I think I will keep the score as is.

---

> > > > > ### Author Response · Authors · 2023-08-18
> > > > >
> > > > > Dear Reviewer aiKz,
> > > > >
> > > > > Thanks very much for your prompt reply!
> > > > > Have a good weekend.
> > > > >
> > > > > Authors of paper #3140

---

### Official Review · Reviewer_ZBA5 · 2023-07-07

**Soundness:** 4 excellent
**Presentation:** 2 fair
**Contribution:** 3 good
**Rating:** 7
**Confidence:** 4

**Summary:**

Using Lie group algebra, this paper derives the general conditions for a network that evolves through a series of states at a continuous set of possible rates.  This would be useful for temporal rescaling, as is frequently observed in motor generation (imagine playing a melody at a different rate than it was learned).  These considerations are applied to develop a continuous attractor network that is time scale equivariant.  Simulations implement a continuous attractor network that drives a set of output neurons to draw a spatial pattern at different rates.

**Strengths:**

The rigor of this approach is admirable.

The inclusion of a continuous attractor network is a nice addition. I'm not aware of a model addressing sequential activation that makes use of this mechanism.

This approach is vastly superior to conventional RNNs, which are ill-suited as a model of neural sequence coding or generation.

**Weaknesses:**

The dependence of this approach on a common zero to the scale ($t_{start}$) is not given sufficient weight.  Reversal of time in this approach (negative alpha, as in fig 1b) requires exchange of the initial condition as well.

Generally speaking, the claims to novelty are overstated.  It is not correct to say that using gain to change the rate of a temporal representation has ``never [been] used in ML studies.'' The entire field of dynamic time warping (e.g., Sakoe and Chiba, 1978) is clearly ML (although not a neural network).  Gutig & Sompolinsky (2009, PLoS Bio) studied pattern classification in a network that exactly modulates the rate of time progression via gain.  More recently there was a workshop paper at NeurIPS last year addressing gain as a control signal in deep networks:
https://memari-workshop.github.io/papers/paper_42.pdf

The Weber law is introduced via a fairly ad hoc mechanism.  Weber law emerges naturally from requiring scale-covariance, rather than scale equivariance.

**Questions:**

What could be the benefit of a time scale-covariant network?  Lindeberg and colleagues (e.g., Lindeberg & Fagestrom, 1996) have studied the conditions for scale-covariant temporal receptive fields for many years.  What is the relationship between this work and Jacques et al., (2022; ICML)?

Would this scale-equivariant network be useful for memory?

It's not clear how useful this approach is generally.  How does the number of patterns that can be generated using a single CAN scale?  What's the architecture one would need to retrieve a large number of such motor patterns?

---

> ### Author Rebuttal · Authors · 2023-08-09
>
> We deeply appreciate your positive review and insightful comments.
>
> > “The dependence of this approach on a common zero to the scale ($t_{start}$) ..."
>
> Our analysis implicitly assumes that the sequences with different speeds start from the same $t_{start}$, and time reversal requires exchanging start timing (see details of $t_{start}$ in Eq. S11). The focus of the present study is the temporal scaling (TS) of sequences rather than temporal alignment between sequences, and the start timing is not critical in temporal scaling. Therefore, we suppress $t_{start}$ in equations to simplify notations. We will explain this more clearly in the revision.
>
> > “Generally speaking, the claims to novelty are overstated...”
>
> Please see *global reply* for our statement about gain-modulation.
>
> Thanks for highlighting those previous studies. We're keen to offer comparisons to underscore our contributions in the revision. Our study stands out for its **theoretical** and **analytical** insight into the representation of TS groups within recurrent network dynamics, which was recognized by Reviewer aiKz. While leveraging gain-modulation to manage TS has precedent (Ref. [2, 6-13], and Maheswaranathan 2019 mentioned by Reviewer tVWd), our analytical approach in identifying the TS group representation is novel. On a closer examination of the papers referenced by the reviewer, we find:
> - Sakoe and Chiba (1978) introduced a warping function to synchronize sequences with varied temporal scales. But as the reviewer rightly pointed out, it doesn't employ neural networks. It is hard to conclude that the gain modulation in the present study was also used in that paper.
> - Gutig 2009 focused on a single neuron mechanism of TS, where the input firing rate modulated the effective time constant of conductance-based integration-and-fire neurons. Even though there's a resemblance between their modulating input firing rate and our gain-modulation, the distinction is that we focused on the recurrent network mechanism of TS, unlike the single neuron mechanism.
> - The NeurIPS Workshop paper interprets the output of the inverse Laplacian transform as neural responses in the network's hidden layer (as illustrated in Eq. 1 and Fig. 1a). They then couple a factor, $\alpha(t)$, with the decaying term, $-sF(s,t)$, which modifies the effective time constant. This is fundamentally different from our gain modulation approach.
>
> Importantly, our research represents a generative model for sequence generation, whereas the mentioned papers focused on the inference of discerning time-warped sequences.
>
> > “The Weber law is introduced via a fairly ad hoc mechanism...”
>
> We agree that Weber’s law, referring to the std of time interval proportional to the mean, satisfies the scale-covariance. The present study does not derive Weber’s law from a normative way because it is not the main focus of the current study. Instead, we only use Weber’s law as a way to demonstrate the biological plausibility of the proposed circuit model, in that Weber’s law has been widely observed in TS experiments in neuroscience research. At the neural circuit level, we found that Weber’s law can be only realized by including the Poisson variability referred to as neuronal spiking variability  (see lines 258-264 in the manuscript), while it will deviate if the injected noise is additive (not presented in the manuscript).
>
> > “What could be the benefit of a time scale-covariant network?...”
>
> We are happy to discuss and compare our study with these papers in the revised manuscript. The time scale-covariant networks mentioned by the reviewer focused on the temporal scale-invariant representation of sequences (e.g., Fig.3C,H in  Jacques 2022), which is beneficial for **recognizing** sequences with different time scales. In contrast, our model aims to investigate the recurrent network mechanism of flexibly **generating** sequences with different temporal scales. For the proposed network model, when inverting the information flow and introducing a feedback loop between the context input and the recurrent network, the proposed network has the potential to recognize (infer) the temporal scaling factors of input sequences.
>
> > “Would this scale-equivariant network be useful for memory?”
>
> Thank you for motivating us to think of another potential function of our proposed neural circuit model, and memory is an important function of the brain. This is an open question and here we provide our thoughts that are yet to be verified by future experiments. The short answer is yes. The hippocampus is essential in forming new memories. A characteristic of hippocampal responses is the theta oscillation, where it was hypothesized theta sequences compress (speed up) the sensory input sequence by about 10 times (Skaggs, J. Neurosci., 1995). It was believed that compressing the temporal sequence by theta oscillation is beneficial for memory formation, and it is possible that the theta oscillation shares a similar computational mechanism with the temporal scale equivariant network. We will discuss this issue in our revised manuscript.
>
> > “It's not clear how useful this approach is generally...”
>
> In the disentangled circuit, the CAN is a "clock" generating a _stereotypical_ neural sequence, and the feedforward circuit represents the pattern. Therefore the number of patterns that can be generated by the CAN is only one, and the number of patterns that can be generated by the whole disentangled network depends on the complexity of the feedforward circuit.
>
> Since the present study focuses on TS in a recurrent network, the feedforward circuit is a toy model where a trained feedforward circuit only generates one pattern, and it needs to be **retrained** to generate another (see line 275). By increasing the complexity of feedforward circuits, e.g., introducing a "gate" to modulate the **functional** feedforward connectivity, the network can generate many sequence patterns (see discussions in lines 330-332).

---

> > ### Comment · Reviewer_ZBA5 · 2023-08-10
> >
> > Thanks for the thoughtful response; my score has increased.

---

> > > ### Author Response · Authors · 2023-08-11
> > >
> > > We appreciate your upgraded score and positive feedback on our paper. Your support inspires us to refine our work further.

---

### Official Review · Reviewer_tVWd · 2023-07-11

**Soundness:** 3 good
**Presentation:** 3 good
**Contribution:** 2 fair
**Rating:** 6
**Confidence:** 3

**Summary:**

The authors present a framework to analyze and realize temporal scaling in networks. Studies in neuroscience have thought about how the brain achieves outputs at different speeds. Here, the authors present a theoretical analysis of temporal scaling group operators, and then posit a disentangled neural circuit that is a recurrent neural network (RNN) feeding to a feedforward network to produce the actual output. The RNN takes in a time scale control input and generates the temporal scaling, while the feedforward circuit produces the pattern for the task. The authors consider an RNN with fixed weights. Finally, simulation results are shown to produce handwritten digits.

**Strengths:**

Originality: While the authors use the previously developed continuous attractor network (CAN), the analyses presented are original. A recurrent circuit that takes in a temporal scaling input and outputs signals at different frequencies has been proposed before in Maheswaranathan, Williams, et al., 2019, which is not cited.

Quality: The paper considers several neuroscience studies and attempts to distill the way neurons encode time from a theoretical perspective, as well as proposes a concrete realization of such a recurrent circuit. Although the concrete nature of the representation may have its advantages to be able to analyze the network, please see weaknesses.

Clarity: The presentation of the paper is very clear.

Significance: The authors formulate a temporal scaling group, but it is unclear whether this leads to additional insight into networks that are able to perform temporal scaling. The recurrent network that the authors posit is a very specific one, where the neurons are a specific function of the scaling, and thus this limits the use of this network (since it cannot be learnt / trained).

**Weaknesses:**

The use of a fixed neural representation of the internal 'time' manifold may be a limitation of the study.

The authors claim that "when generating movements with different time scales, a recurrent circuit's responses u(t) evolve along the same low-dimensional manifold with different speeds", and "overlap perfectly". However, in most studies, the responses are in fact separated in state space. Also see Saxena, Russo, et al., 2022. In fact, even in the proposed paper, it seems that the neurons' responses are separated in PC space (Fig 3C). Please clarify.

The simulation experiments are extremely limited - the authors only consider simple x and y coordinates of hand-written digits, and only show their results for the digit '6'. However, they claim that the responses can be mapped to an arbitrary, complex sequence. Please expand.

It is unclear what we learned from this paper. The theoretical analysis formulates a temporal scaling group, but the takeaways are limited. The circuit proposed by the authors can implement temporal scaling, but so can other circuits (since the authors consider a hand-crafted network). It would be helpful if the authors can extend their results to arbitrary or trained networks.

**Questions:**

Please see weaknesses above.

**Limitations:**

The authors have adequately addressed limitations of their work.

---

> ### Author Rebuttal · Authors · 2023-08-08
>
> Thank you for the reviewer’s comments, especially the appreciation of our theoretical derivations in identifying how temporal scaling group operators are represented in the recurrent network dynamics.
>
> > “Originality: While the authors … which is not cited.”
>
> Thank you for directing us to this paper. We are happy to cite it in our revised manuscript. It aptly supports and provides motivation for our network model that receives a temporal scaling (TS) control input. Notably, we've also observed that the dimensionality reduction analysis in our network aligns closely with the phenomena presented in Maheswaranathan, 2019.
>
> While the continuous attractor network stands as a canonical model in neuroscience, its application to TS remains largely unexplored, as acknowledged by the Reviewer ZBA5. Numerous previous studies have delved into training a network to achieve TS (highlighted on line 26 of our manuscript). However, there remains a large gap in the theoretical analysis of the dynamics of such recurrent networks. One of the standout contributions of our study is the theoretical and analytical insight into how the abstract TS group operators manifest in recurrent circuit dynamics (as discussed in lines 231 – 243), which has not been documented before.
>
> We believe that these theoretical insights into the abstract group representation within concrete network dynamics will greatly enhance our comprehension of the algebraic structure of network dynamics. Further, it sets the foundation for the development, learning, and analysis of network representations for even more intricate group structures in future research.
>
> > “The use of a fixed neural representation… a limitation of the study.”
>
> Understanding the general principle of "time" representation in neural circuits remains challenging. It's not yet fully clear whether neural representations of "time" are fixed or adaptive based on context. Our proposed neural circuit model is rooted in a recurring observation: the presence of a one-dimensional "time" manifold in the neural population response (Refs. [2-7]). This observation guided our construction of a neural population and its dynamics around this one-dimensional "time" manifold.
> One notable advantage of having a fixed "time" representation is the ability to disentangle "time" from sequence patterns (Fig. 2A), which streamlines the joint representation of sequence speeds and patterns.
> While the contextually flexible neural representation of "time" is enticing, modeling such a neural circuit lacks robust experimental backing. Tackling this challenge will be an exciting endeavor in the future, but it is likely beyond the scope of our current research.
>
> > “The authors claim … separated in PC space (Fig 3C).”
>
> Thank you for mentioning Saxena, et al., 2022. We found our network model exhibits **consistent** results with this paper and we are happy to cite it in our revised manuscript to support our model.
> The “perfect overlap” in the original paper means the trajectories will overlap perfectly only in the **“time” manifold**, as shown in the bottom panel in Fig. 3C. This result is consistent with the result reported in Fig. 5C in Ref. 6, and is similar with the Fig. 3d in Saxena, et al., 2022 (our “time” manifold corresponds to the gray circles in the Fig. 3d). Indeed, in the proposed recurrent circuit model, the trajectories with different speeds are **separated** along other dimensions. Fig. 3C, top panel shows the trajectories are separated along the direction of the input gain $\alpha$. We will revise the text to make it clear.
>
> > “The simulation experiments are extremely limited…”
>
> In the disentangled neural circuit model, the feedforward circuit maps the sequential activities of the RNN to a specific temporal sequence.  We used handwritten digits to demonstrate this ability, which was widely used in neuroscience studies (Ref. 10-11).  Due to the page limit, we only presented the sequence of digit ‘6’ in the main text.  We do have the results of generating the sequence of other digits from 0 to 10, which were presented in the supplementary Fig.S3. Nonetheless, to use the proposed disentangled circuit model to generate new sequence patterns, e.g., other digits, the feedforward circuit needs to be retrained (see the discussion in the original manuscript, lines 328-331). We admit that the feedforward circuit might be over-simplified, since our main focus is on the TS in the recurrent circuit. Looking ahead, we are keen on enhancing the feedforward circuit's capabilities, allowing it greater flexibility in producing diverse sequence patterns, as suggested in lines 330-332 of the manuscript.
>
> > “It is unclear what we learned from this paper ...”
>
> Please see *global reply* for the contributions of our work.
> A novel takeaway is the **analytical** results of the TS group representation in the nonlinear recurrent neural circuit dynamics (lines 231-243). The TS recurrent circuit models have been developed and learned (ref. [2, 6-13], and Maheswaranathan, et al., 2019), but no previous result can yield an analytical result identifying the representation of the abstract group operators in the circuit dynamics. Moreover,  our work proposes a continuous attractor network to implement the TS group for the first time, which is also confirmed by the Reviewer ZBA5. In contrast, most previous studies trained chaotic RNNs to implement TS transformations.
>
> The inclusion of a hand-crafted network model in the present study is to obtain analytical results, and otherwise, it is unlikely to get. We are interested in using the theoretical insight obtained from the present study to design a new paradigm to train RNNs to implement TS.

---

> > ### Author Response · Authors · 2023-08-19
> >
> > Dear Reviewer tVWd,
> >
> > Thanks for reviewing our paper!
> >
> > We are wondering whether our reply addresses your concerns. If you have other concerns, please let us know immediately and we are happy to explain.
> >
> > Thanks again!
> > Authors of paper #3140

---

> > > ### Comment · Reviewer_tVWd · 2023-08-21
> > >
> > > Thank you for your detailed rebuttal and for the clarifications. I have increased my rating accordingly.

---

### Author Rebuttal · Authors · 2023-08-08

We sincerely appreciate all reviewers' comments! Here we reply to some common questions raised by reviewers.

### Novelty and contribution
As some reviewers asked whether Lie group theory is necessary for this study, we emphasize the objectives and contributions of our work that include **three** parts:
1. Our study provides **theoretical** and **analytical** understanding of temporal scaling (TS) group representation in the biologically plausible recurrent network dynamics (lines 231 – 243). Specifically, we start from the Lie group to derive the TS operations on neural representations (Eqs. 7-10), and then compare it with the theoretical analysis of network dynamics (Eq. 23) to identify the TS group operator representation. Nearly no previous study analytically link the TS group with a nonlinear recurrent neural circuit dynamics.
1. We rigorously derive the continuous attractor dynamics based on the Lie group framework, which is a novel implementation of TS transformation (acknowledged by the Reviewer ZBA5).
1. We rigorously introduce group equivariance to define temporal scaling.  Equivariance has become a trending topic in machine learning (Cohen & Welling, 2016) and has been widely used in, e.g., geometric machine learning,  molecular design (Hoogeboom, 2022), fluid dynamics (Wang, 2020), particle physics (Bogatskiy, 2020), etc. We believe using the Lie group to understand the cognitive and neural computation in the brain would be fundamental in neuroscience research.

Other new results include
1. A negative gain can induce “time reversal” which has not been reported to our best knowledge.
1. Disentanglement between time control and pattern generation, which possibly corresponds to the mismatch of neural and muscle activities that were found by Saxena, et al., 2022, mentioned by the Reviewer tVWd.

Combined, to the best of our knowledge, the present study is one of the earliest studies that link 1) the abstract group operators, 2) biologically plausible recurrent network dynamics, and 3) neurobiological experiments together. Most earlier studies only bridged the above aspects of 1) and 2), or 2) and 3), but not all three of them.  We believe having such theoretical results of abstract group representation in concrete network dynamics would greatly gain our understanding of the algebraic structure of network dynamics, and can help us build/learn network representation of more complicated group structures in the future.

### Gain modulation
The proposed neural circuit model receives a time-invariant control input whose gain can be modulated to change the temporal scale of the sequences in the recurrent circuit. The **gain-modulation** is a widely accepted neuroscience operation that refers to only modulating the **magnitude** of inputs, e.g., multiplying the control input $I(x|z_\infty)$ (vector) by the gain $\alpha$ (scalar) (see Eq. 23). Indeed, using gain-modulation to control temporal speed has been used before, e.g., previous studies trained recurrent networks that receive a gain-modulated temporal scaling input (Ref. [2, 6-13], and Maheswaranathan, et al., 2019 mentioned by the Reviewer tVWd). However, no previous study **theoretically** identified the control input gain represents the temporal scaling factor as reported in the present study. The novelty of our method was also acknowledged by the Reviewer aiKz in his/her statement of originality about our work.

---

### Decision · Program_Chairs · 2023-09-21

**Decision:**

Accept (poster)

**Comment:**

The reviewers appreciated the novelty, originality and clarity of this manuscript, and I congratulate the reviewers for their careful and detailed rebuttals, which were helpful for clarifying the paper's theoretical contributions. Although several reviewers felt that the Lie group theory framework was not a necessary component to the manuscript's findings, the overall sentiment among reviewers was that the manuscript's contributions (e.g., continuous attractor implementation of temporal scaling, reproduction of biological findings) were novel and important enough to warrant acceptance.  Thus, I'm pleased to report that this paper has been accepted to NIPS.  Congratulations!  Please revise the manuscript to address all reviewer comments and questions.